# Hormone Regulation Effect of Blue Light on Soybean Stem Internode Growth Based on the Grey Correlation Analysis Model

**DOI:** 10.3390/ijms26094411

**Published:** 2025-05-06

**Authors:** Chang Wang, Shuo Huang, Baiyang Yu, Fuxin Shan, Xiaochen Lyu, Chao Yan, Chunmei Ma, Baiwen Jiang

**Affiliations:** 1College of Agriculture, Northeast Agricultural University, Harbin 150030, China; 18249555621@163.com (C.W.);; 2School of Resources and Environment, Northeast Agricultural University, Harbin 150030, China

**Keywords:** *Glycine max* (soybean), blue light, stem internode, hormones ratio, proteomics, grey correlation analysis

## Abstract

Blue light serves as a critical environmental cue regulating *Glycine max* (soybean) stem morphology, yet the hormonal mechanisms underlying varietal differences remain unclear. Previous studies have highlighted the role of blue light in modulating plant architecture, but the specific hormone interactions driving morphological divergence between soybean varieties remain underexplored. Two soybean varieties with contrasting stem phenotypes—Henong 60 (HN60, tall) and Heinong 48 (HN48, dwarf)—were subjected to 0% (full light) and 30% (shade) transmittance conditions, supplemented with blue light (450 nm, 45.07 ± 0.03 μmol·m^−2^·s^−1^). Stem anatomical traits (xylem area, cell length), hormone profiles, and proteomic changes were analyzed. Grey correlation analysis quantified relationships between hormone ratios and plant height. Blue light increased soybean stem xylem area and diameter while reducing plant height and cell longitudinal length. This treatment concurrently reduced growth-promoting hormones (gibberellin A_3_ (GA_3_), indole-3-acetic acid (IAA), brassinolide (BR)) and increased growth-inhibiting hormones (salicylic acid (SA), jasmonic acid (JA), strigolactones (SLs)), thereby inhibiting stem elongation. Although exogenous GA_3_ promoted hypocotyl elongation, it failed to counteract blue-light-induced inhibition. Proteomic analysis identified 16 differentially expressed proteins involved in hormone signal transduction pathways. Grey correlation analysis highlighted cultivar-specific hormone ratio impacts: GA_3_/JA, GA_3_/SA, and BR/SLs significantly influenced HN60 plant height, while GA_3_/SLs, IAA/SLs, and BR/SLs were critical for HN48, demonstrating highly significant positive correlations. The differential sensitivity of growth-promoting/inhibiting hormone ratios to blue light drives varietal morphological divergence in soybean stems. This study establishes a hormonal regulatory framework for blue-light-mediated stem architecture, offering insights for crop improvement under light-limited environments.

## 1. Introduction

Light serves as a critical environmental regulator governing plant morphogenesis and differentiation [1,2,3,4,5,6]. Among spectral components, blue light constitutes a pivotal photomorphogenetic signal exhibiting species-specific effects on developmental processes [7,8]. Physiological studies demonstrate that elevated blue light intensity reduces cucumber seedling growth and plant height through diminished cell wall extensibility [9,10]. This inhibitory mechanism contrasts with findings in *Glycine max* (soybean), where blue-light-mediated suppression of internode elongation primarily results from reduced epidermal cell dimensions and decreased cell density per internode unit [1,6,11].

Phytohormones function as essential signaling mediators in plant responses to spectral quality [12]. Gibberellins (GAs) and indole-3-acetic acid (IAA) coordinately regulate shade-avoidance responses and elongation growth [13,14,15]. Both reduced red/far-red ratios and low-intensity blue light environments elevate endogenous IAA and GA levels, thereby promoting stem elongation [16,17]. The inhibitory effect of blue light on petunia main stem elongation has been mechanistically linked to GA homeostasis modulation [18], while blue light exposure also influences abscisic acid (ABA), salicylic acid (SA), and IAA accumulation patterns [19]. Exogenous GA_3_, IAA, and 24-epibrassinolide application significantly enhances soybean hypocotyl elongation under white light, whereas biosynthetic inhibitors counteract shade-induced elongation, suggesting GA_3_ operates downstream of auxin and brassinosteroid signaling pathways [20]. Crucially, plant developmental outcomes emerge from hormonal crosstalk rather than isolated hormone actions. Ratios between hormonal groups, including GA_3_/IAA, GA_3_/cytokinins (CKs), and GA_3_/ABA, demonstrate regulatory significance in adventitious root formation in passion fruit cuttings [21]. Similarly, elevated IAA/ABA and CK/ABA ratios promote *Pinus tabulaeformis* vegetative growth [22], while reduced ABA/IAA ratio impairs tomato growth vigor [23]. Notably, the IAA/gibberellin A1 (GA_1_) ratio has been proposed as a key indicator of soybean photomorphogenetic responses, with far-red supplementation or reduced light intensity differentially modulating this ratio across leaf, petiole, and stem tissues [24]. However, current research predominantly focuses on individual hormone effects, with limited exploration of the synergistic/antagonistic relationships between hormonal ratios—a critical knowledge gap that this study addresses regarding blue light regulation of soybean stem elongation.

Advances in proteomics have illuminated plant photomorphogenetic mechanisms under varying light spectra. Comparative proteomic analysis revealed blue-light-induced suppression of potato seedling growth through differential regulation of hormone signaling proteins and concomitant GA_3_/IAA reduction [25]. Contrastingly, blue light enhances industrial hemp growth via auxin biosynthesis upregulation and starch accumulation [26]. Grey-system-theory-based correlation analysis provides a robust framework for multi-parameter optimization in complex biological systems [27]. While widely applied in crop studies evaluating abiotic stress responses [28,29,30,31,32,33,34], its potential in photomorphogenetic research remains underexplored. This study utilized two contrasting soybean cultivars, Henong 60 (HN60) and Heinong 48 (HN48), characterized by distinct morphological architectures and lodging resistance capacities, as experimental models. Employing integrated correlation and grey relational analyses, we systematically identified the core phytohormones governing soybean growth modulation under blue light. A multi-omics approach combining morphological phenotyping, histological characterization, hormonal quantification, gene expression profiling, and proteomic interrogation revealed coordinated hormonal networks underlying blue-light-mediated stem elongation dynamics. These findings establish a mechanistic framework for photomorphogenetic regulation in legumes, providing both theoretical insights into light-quality adaptation and potential applications for precision cultivation.

## 2. Results

Comparative analysis revealed distinct stem anatomical characteristics between treatments (Figure 1). Shade + blue light (SB) treatment plants exhibited enhanced stem diameter and xylem area compared to shading (S) treatment counterparts. Both treatments displayed progressive expansion of medullary cells from peripheral to central regions, with SB treatment specimens showing improved radial cell alignment and reduced longitudinal cell dimensions.

Figure 2 demonstrates blue-light-mediated suppression of internode elongation, particularly evident in HN60′s fourth (44.42% reduction) and fifth internodes (34.32% reduction), contrasting with HN48′s broader inhibition across the third to sixth internodes (19.95–58.56% reduction). These findings indicate blue light’s dual regulatory role in stem thickening and longitudinal growth restriction.

Temporal analysis of hypocotyl elongation revealed biphasic growth patterns (Table 1). Both cultivars exhibited rapid initial elongation (0–3 days), followed by growth stabilization (3–7 days). Blue light significantly suppressed hypocotyl elongation in both varieties, with treatment differences reaching statistical significance from day 3 onward. Conversely, plant height increased by 64.02% (HN60) and 64.64% (HN48) under blue light, demonstrating spectral-quality-dependent growth modulation.

Table 2 illustrates the significant impact of blue light on stem hormone profiles in both cultivars. Relative to the S treatment, the SB treatment induced a significant reduction in GA_3_ and IAA concentrations across the third, fourth, and fifth internodes of HN60 and HN48. Stem BR content was also significantly decreased under SB treatment, with the exception of the third internode, suggesting that blue light exposure suppresses growth-promoting hormone biosynthesis. Conversely, SB treatment resulted in marked increases in JA, SA, and SLs concentrations in all evaluated internodes, indicating a potential role of blue light in enhancing stress-responsive hormone accumulation.

As presented in Table 3, compared to the darkness, BL treatment elicited significant reductions in GA_3_, IAA, and BR levels while promoting substantial increases in JA, SA, and SLs concentrations in both cultivars. These findings demonstrate that supplementary blue light under dark conditions can significantly modulate phytohormone concentrations, particularly by attenuating the accumulation of growth-promoting hormones GA_3_, IAA, and BR.

Analysis of Figure 3 reveals that relative to distilled water treatment, application of 100 mg L^−1^ GA_3_ significantly promoted soybean hypocotyl elongation, with the most pronounced changes occurring within 0–3 days post-treatment. Hypocotyl length under blue light conditions exceeded that observed in darkness, and HN48 exhibited more pronounced hypocotyl length variations compared to HN60. These results indicate that 100 mg L^−1^ GA_3_ effectively promotes hypocotyl elongation, yet fails to attenuate blue-light-induced inhibition of this process. Exogenous application of 100 mg L^−1^ IAA, 10 mg L^−1^ IAA, and 100 mg L^−1^ BR suppressed hypocotyl elongation in both cultivars under darkness. Notably, BR supplementation under blue light significantly augmented hypocotyl length in HN48 at 2, 3, and 4 days, whereas IAA had no measurable effect. HN48 demonstrated higher sensitivity to IAA, as reflected by greater hypocotyl length alterations.

Plant hormone signaling pathways play a pivotal role in soybean stem development. This study identified 16 differential expressed proteins associated with phytohormone signal transduction under blue light treatment compared to darkness (Table 4). These proteins were involved in salicylic acid, gibberellin, jasmonic acid, and auxin metabolic and signal pathways. Blue light treatment upregulated nonexpressor of pathogenesis-related genes 1 (NPR1) in the SA signal pathway and acetyl-CoA acyltransferase 1 (ACAA1), lipoxygenase 1_5 (LOX1_5), lipoxygenase2S (LOX2S), and hydroperoxide lyase (HPL) in the jasmonic acid metabolic pathway, while downregulating gibberellin receptor GA-INSENSITIVE DWARF 1 (GID1C), allene oxide cyclase (AOC), and allene oxide synthase (AOS) in jasmonic acid metabolism and aldehyde dehydrogenase (ALDH) in auxin metabolism (Table 4).

Figure 4 illustrates significant negative correlations between HN60 plant height and GA_3_/IAA and BR/IAA ratios, alongside significant positive correlations with GA_3_/SA, GA_3_/JA, GA_3_/SLs, IAA/GA_3_, IAA/BR, IAA/SA, IAA/JA, IAA/SLs, BR/SA, BR/JA, and BR/SLs ratios. HN48 exhibited negative correlations with GA_3_/BR and extreme negative correlations with IAA/BR, while showing extreme positive correlations with GA_3_/SA, GA_3_/JA, GA_3_/SLs, IAA/SA, IAA/JA, IAA/SLs, BR/IAA, BR/SA, BR/JA, and BR/SLs. These results highlight the close association between plant height and hormone ratios in both cultivars.

Grey correlation analysis, shown in Figure 5, identifies the top three indices most strongly correlated with HN60 plant height as GA_3_/JA, GA_3_/SA, and BR/SLs, whereas GA_3_/IAA, BR/IAA, and GA_3_/BR showed weaker correlations. For HN48, the highest correlations were observed with IAA/SLs, GA_3_/SLs, and BR/SLs, while IAA/BR, GA_3_/BR, and BR/SA displayed lower associations. These findings underscore the differential effects of hormone interactions on plant height and significant cultivar-specific responses.

## 3. Discussion

Shading promotes internode elongation through modifications to soybean stem anatomical structure [35]. Liu et al. [36] reported that shading significantly reduced stem xylem area, pith area, and xylem proportion while increasing pith proportion in soybean, consistent with the findings of Wen et al. [37], who observed reduced xylem and pith areas under intensified shading. In tomato seedlings, Falcioni et al. [19] demonstrated that blue light increased cortical parenchyma cell thickness and number while decreasing xylem cell count compared to white light. Our results align with these observations, showing that blue light increased stem xylem area but reduced cell longitudinal length, thereby inhibiting soybean hypocotyl elongation, as previously reported by Wang et al. [6]. Cryptochrome-mediated blue light signaling is known to suppress hypocotyl elongation [38,39], with Arabidopsis *cry1-304* mutants displaying exaggerated hypocotyl growth under blue light [40]. The basic leucine zipper (bZIP) transcription factor ELONGATED HYPOCOTYL 5 (HY5), a key photomorphogenesis regulator [41], was shown by Alabadi et al. [42] to interact with GAs signaling pathways. Our study detected upregulated cry1a and HY5 expression under blue light (Appendix A), suggesting their involvement in mediating hypocotyl growth responses in soybean.

A diverse array of phytohormones participate in the shade avoidance signaling cascade. Under shading stress, soybean stems exhibit reduced ABA and zeatin concentrations, concurrent with elevated IAA and GA_3_ levels. Increasing shading intensity correlates with enhanced internode elongation, accompanied by rising GA_3_ and SA contents and decreasing ABA concentration [3,43]. These trends align with the current study’s observations. Blue light treatment significantly reduces growth-promoting hormones—GA_3_, IAA, and BR—while augmenting stress-responsive hormones—SA, JA, and SLs. Phytochrome-interacting factors (PIFs) act as negative regulators of photomorphogenesis and serve as critical integrators between light and hormonal signaling pathways [44]. Hornitschek et al. [45] demonstrated that phytochrome-interacting factor 4 (PIF4) and phytochrome-interacting factor 5 (PIF5) modulate Arabidopsis seedling growth through direct regulation of auxin biosynthesis genes and signaling components. In this study, blue light exposure downregulated the relative expression levels of *PIF4A* and *PIF4B* (Appendix A), suggesting that blue light may influence hormone concentrations via transcriptional regulation of PIFs.

GA_3_ plays a pivotal role in regulating cell elongation [46], specifically governing the longitudinal cell expansion [47]. The soluble receptor GA-INSENSITIVE DWARF1 (GID1) perceives gibberellin signals [48], while DELLA proteins act as repressors of the gibberellin signaling pathway [49]. In this study, blue light exposure downregulated the gibberellin receptor protein GID1c (Table 4). Concurrent with this, *DELLA* gene expression and GA catabolic gene *GA2ox4* were upregulated, whereas GA biosynthetic gene *GA20OX1* was downregulated (Appendix A), aligning with findings from Zhao et al. [50]. These results indicate that blue light inhibits GA biosynthesis and promotes its degradation, thereby suppressing hypocotyl elongation. Exogenous application of 100 mg L^−1^ GA_3_ under darkness maximized soybean hypocotyl elongation. However, while GA_3_ supplementation under blue light increased hypocotyl length compared to blue light alone, it remained shorter than dark-grown controls (Figure 3), confirming that GA_3_ cannot fully reverse blue-light-induced growth inhibition. IAA, essential for plant elongation and development [51], is synthesized from indole-3-acetylaldehyde via aldehyde dehydrogenase (ALDH) in *Escherichia coli* [52]. Here, downregulation of ALDH in the tryptophan metabolic pathway (Table 4) suggests impaired IAA biosynthesis under blue light. Yang et al. [53] reported dose- and time-dependent stem elongation in pea seedlings treated with exogenous IAA. In contrast, our study observed that 10 mg L^−1^ and 100 mg L^−1^ IAA significantly inhibited soybean hypocotyl growth in darkness. While IAA-treated hypocotyls under blue light showed marginal increases compared to blue light alone, these differences were not statistically significant (Figure 3), highlighting species-specific responses to IAA and the need for further dose-response investigations under blue light. BR is a key regulator of plant elongation [54]. Exogenous BR promotes stem and leaf elongation in sweet pepper [55] but inhibits growth in *Sophora davidii* at high concentrations (50 mg L^−1^) [56]. Our results corroborate this duality: 100 mg L^−1^ BR suppressed hypocotyl growth in darkness but enhanced elongation under blue light (Figure 3), indicating that BR-mediated growth regulation is dose- and light-dependent.

Additionally, SA inhibits Arabidopsis hypocotyl elongation, with NPR1 serving as the central regulator of SA signaling [57]. Blue light exposure upregulated NPR1 protein abundance and increased SA content in this study. Lipoxygenase (LOX), the first committed enzyme in JA biosynthesis, initiates JA production by catalyzing the oxygenation of polyunsaturated fatty acids to linoleic and linolenic acids [58]. HPL, a downstream enzyme in the LOX-dependent lipid oxidation pathway, cleaves LOX-generated hydroperoxides into short-chain aldehydes and ω-oxo-acids [59]. ACAA1 participates in fatty acid metabolism by catalyzing the final step of the β-oxidation pathway, contributing to fatty acid elongation and degradation [60]. Upregulation of LOX, HPL, and ACAA1 under blue light treatment in this study suggests enhanced JA biosynthesis.

Phytohormone interactions and homeostasis play critical roles in plant morphogenesis and development. Wang et al. [61] reported that higher IAA/GA_3_ in peony (*Paeonia suffruticosa*) roots promoted adventitious root induction and elongation. Hou et al. [62] demonstrated that IAA/ABA and IAA/CTK ratios positively correlate with rooting efficiency in Chinese chestnut (Microshoots of *Castanea mollissima*). Similarly, Quan et al. [63] found that increased IAA/ABA and IAA/ZR ratios facilitated root primordium formation and early rooting in *Catalpa bignonioides* cuttings. Collectively, these studies highlight the context-dependent regulatory effects of hormone interactions on plant growth. In this study, grey correlation analysis revealed cultivar-specific hormone ratio impacts on plant height. GA_3_/JA, GA_3_/SA, and BR/SLs ratios exerted significant effects on HN60, whereas IAA/SLs, GA_3_/SLs, and BR/SLs ratios were critical for HN48. Blue light significantly reduced these key ratios in both cultivars (Appendix A), indicating that differential balances between growth-promoting and growth-inhibiting hormones contribute to morphological divergence.

## 4. Materials and Methods

### 4.1. Experimental Design

Two soybean cultivars with contrasting lodging resistance and morphological traits were selected: Henong 60 (dwarf lodging-resistant variety) and Heinong 48 (high-stalk lodging-susceptible variety).

#### 4.1.1. Experiment I: Shading and Blue Light Supplementation

A pot experiment was conducted using cylindrical pots (33 cm diameter × 30 cm height, 1 cm drainage hole diameter) filled with 15 kg soil. Fertilization included 1.5 g·pot^−1^ diammonium hydrogen phosphate (N: 18%, P_2_O_5_: 46%) and 0.75 g·pot^−1^ potassium sulfate (K_2_O: 50%). Seeds were sown in a single row along the pot diameter (8 holes·pot^−1^, 2 seeds·hole^−1^), thinned to 4 seedlings·pot^−1^. Treatments were applied at the V_3_ growth stage until V_5_:

S: shading with a black net (30% light transmittance).

SB: shading + supplementary blue light (LED tubes, 12 W, uniform lamp bead arrangement).

LED tubes were positioned 1 cm above the soybean canopy and adjusted vertically as plants grew. Irradiation occurred continuously from sunrise to sunset. At harvest, internode length and plant height were measured. Stem segments (third, fourth, fifth internodes) were rinsed with PBS buffer, wrapped in aluminum foil, snap-frozen in liquid nitrogen, and stored at −80 °C for hormone analysis. Anatomical observations of the third internode were performed.

#### 4.1.2. Experiment II: Controlled Environment Chamber Study

A closed light chamber (120 cm × 60 cm × 180 cm) was used with 9 cm diameter × 10 cm height pots filled with flower nutrient soil. Uniform seeds were sown in three holes·pot^−1^ (2 seeds·hole^−1^), thinned to 1 seedling·pot^−1^. Seedlings were grown in darkness at 25 °C until hypocotyls reached ~5 cm. Blue light treatment (LED tube, 45.07 ± 0.03 μmol·m^−2^·s^−1^) was applied unilaterally 1 cm from the stem. During the whole experiment, the temperature of the incubator was maintained at 25 ± 1 °C, and the relative humidity was controlled at 50 ± 5%. Hypocotyl length and plant height were measured at 1, 3, 5, and 7 days post-treatment. Hypocotyl segments were collected at 3 days for hormone quantification, gene expression, and proteomic analysis. Each treatment was repeated three times.

#### 4.1.3. Experiment III: Exogenous Hormone Treatments

Under the same sowing and growth conditions as Experiment II, the seedlings were cultured to about 5 cm hypocotyls at 25 °C in the dark, and then the soybean hypocotyls were treated with light and exogenous hormones. The specific settings are as follows: (1) Four treatments: darkness + distilled water, darkness + GA_3_, blue light + distilled water, and blue light + GA_3_. The concentration of GA_3_ solution (GA_3_ purity HPLC ≥ 90.0%, purchased from Beijing Solarbio Science & Technology Co., Ltd. (Beijing, China)) was 100 mg L^−1^. (2) Four treatments: darkness + distilled water, darkness + IAA, blue light + distilled water, and blue light + IAA. The concentration of IAA solution (IAA purity ≥ 98%, purchased from Beijing Solarbio Science & Technology Co., Ltd. (Beijing, China)) was 100 mg L^−1^ and 10 mg L^−1^. (3) Four treatments: darkness + distilled water, darkness + BR, blue light + distilled water, blue light + BR 4 treatments, in which BR solution (24-epibrassinolide purity ≥ 90%, purchased from Beijing Solarbio Science & Technology Co., Ltd. (Beijing, China)). The solution concentration was 100 mg L^−1^. Each treatment received 1 mL solution. Hypocotyl length was measured at 0, 1, 2, 3, and 4 days. Light treatment parameters were identical to Experiment II.

### 4.2. Determination Indicators and Methods

#### 4.2.1. Determination of Internode Length and Plant Height

The length of each internode of soybean stem and the length from soybean cotyledon scar to growth point were measured using a ruler.

#### 4.2.2. Observation of Anatomical Structure

Soybean plants of uniform growth were carefully selected. Using a sharp blade, the middle part of the stem was cut into 5 mm segments. These segments were promptly placed into a brown bottle filled with FAA fixative. A vacuum pump was employed to evacuate the air from the samples until they sank to the bottom of the container. Subsequently, the fixed samples underwent a series of treatments. First, they were dehydrated to remove moisture and then made transparent to facilitate better visualization. The sections were then stained with toluidine blue. After staining, the samples were embedded in paraffin wax. The wax-embedded samples were sliced into thin sections, which were then mounted onto glass slides. The slides were baked to ensure proper adhesion of the sections. Next, the paraffin wax was removed through a dewaxing process. After dewaxing, the sections were re-stained and finally sealed to preserve the samples. The prepared sections were observed under a positive optical microscope (Nikon Eclipse E100). An imaging system (Nikon DS-U3) was used to capture high-quality photographs of the sections for further analysis.

#### 4.2.3. Hormone Concentration Quantification

The concentrations of GA_3_, IAA, BR, JA, SA, and SLs were measured using enzyme-linked immunosorbent assay (ELISA) kits (purchased from Shanghai Enzyme-linked Biotechnology Co., Ltd. (Shanghai, China)). Briefly, 0.1 g of tissue samples was homogenized in 0.9 mL phosphate buffer solution (pH 7.4). After centrifugation at 12,000× *g* for 15 min at 4 °C, supernatants were collected. The ELISA protocol followed the manufacturer’s instructions: sample addition → enzyme conjugate addition → incubation (37 °C, 30 min) → washing (×5) → chromogenic substrate addition (37 °C, 15 min) → stop solution addition. Absorbance was measured at 450 nm using a microplate reader.

#### 4.2.4. Detection of Real-Time Quantitative PCR Analysis

In this experiment, *Actin* served as the internal reference gene, and gene-specific primers were designed using Primer Premier 5.0 software. Relative expression levels of photoreceptors, phytochrome-interaction factors, HY5 transcription factor, gibberellin synthesis and decomposition gene, and DELLA protein gene were quantified by real-time quantitative polymerase chain reaction (RT-qPCR). The detailed detection steps are described in Appendix A.

#### 4.2.5. Proteomic Analysis

After 3 days of dark and blue light treatment, 3 g of soybean hypocotyls were collected and divided into three parts, rinsed with PBS buffer, wrapped in aluminum foil, snap-frozen in liquid nitrogen, and stored at −80 °C. Proteome analysis was performed by Shanghai Applied Protein Technology Co. Ltd. according to the method of Wang et al. [5]. Three technical replicates were included for each sample. Specific steps are outlined in Appendix A.

#### 4.2.6. Correlation Analysis

Pearson correlation coefficient was used to calculate the correlation between plant height and hormone ratio of the two varieties, and R Studio (https://posit.co/download/rstudio-desktop/, accessed on 28 April 2025) was used for mapping.

#### 4.2.7. Grey Correlation Analysis Model

Firstly, the data are dimensionlessly processed according to Formulas (1) and (2):(1)$$Y_{i}=\frac{X_{i}(k)}{\frac{1}{m}\sum_{k=1}^{m}X_{i}(k)}$$

In the formula, *Y_i_* is the mean value sequence of each sub-factor; *X_i_* (*k*) is a sub-factor sequence, where *i* denotes the hormone ratio (*i* = 1, 2, 3, …, 16); and *k* is the number of data (*k* = 1, 2, 3, …, 18).(2)$$Y_{o}=\frac{X_{o}(k)}{\frac{1}{m}\sum_{k=1}^{m}X_{o}(k)}$$

In the formula, *Y_o_* is the mean value sequence of the parent factor; *X_o_* (*k*) is the maternal factor sequence, that is, soybean plant height; and *k* is the number of data (*k* = 1, 2, 3, …, 18).

The calculation of correlation coefficient (*k*) is as follows: the number of parent factors is listed as {*Y_o_* (*k*)}, and the number of sub-factors is listed as {*Y_i_* (*k*)}. In *k*, the grey correlation coefficient between {*Y_i_* (*k*)} and {*Y_o_* (*k*)} is:(3)$$\delta_{oi}(k)=\frac{{{△}_{min}+\rho △}_{max}}{{{△}_{0i}(k)+\rho △}_{max}}$$

In the formula, the absolute difference of two sequences at *k* points, that is, $\delta_{oi}(k)$ = |*Y_o_* (*k*) − *Y_i_* (*k*)|; Δ*_min_* and Δ*_max_* are the maximum and minimum values of absolute difference, respectively; and *ρ* is the resolution coefficient, which is 0.5 in most cases.

The calculation formula of grey correlation degree (γ_oi_) is as follows:(4)$$\gamma_{oi}=\frac{1}{m}\sum_{k=1}^{m}\delta_{oi}(k)$$

### 4.3. Statistical Analysis

Data analysis and graphs were performed using SPSS 21.0 and Microsoft Excel 2010 software. The proteins with significantly different expression were screened according to the fold change > 1.2 times (upregulation by more than 1.2 fold or downregulation of less than 0.83 fold) and *p*-value < 0.05 (*t*-test) [5].

## 5. Conclusions

Blue light treatment promoted soybean stem thickening by increasing xylem area and reducing cell longitudinal length. This treatment also reduced growth-promoting hormones (GA_3_, IAA, and BR), and increased growth-inhibiting hormones (SA, JA, and SLs), thereby inhibiting stem internode elongation and reducing plant height. Proteomics analysis identified 16 differentially expressed proteins involved in salicylic acid, gibberellin, jasmonic acid, and auxin metabolic and signaling pathways. Exogenous GA_3_ promoted hypocotyl elongation in darkness but failed to alleviate blue-light-induced inhibition. Exogenous IAA and BR suppressed hypocotyl growth in darkness but had minimal effects under blue light. Grey correlation analysis revealed cultivar-specific hormone ratio impacts: GA_3_/JA, GA_3_/SA, and BR/SLs were strongly positively correlated with HN60 plant height, while GA_3_/SLs, IAA/SLs, and BR/SLs were critical for HN48. These results suggest that differential balances between growth-promoting and growth-inhibiting hormones contribute to morphological divergence between the two soybean varieties.

## Figures and Tables

**Figure 1 ijms-26-04411-f001:**
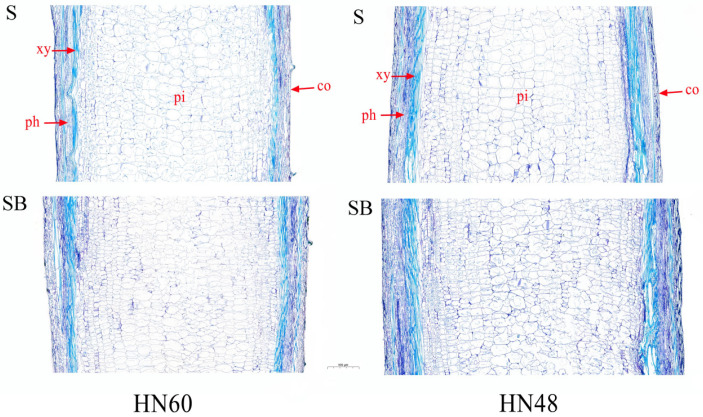
Anatomical structure of stem longitudinal section under blue light treatment and under shading. HN60: Henong 60; HN48: Heinong 48. S: shading treatment; SB: shade + blue light. co: cortex; xy: xylem; ph: phloem; pi: pith. The scale is 500 μm.

**Figure 2 ijms-26-04411-f002:**
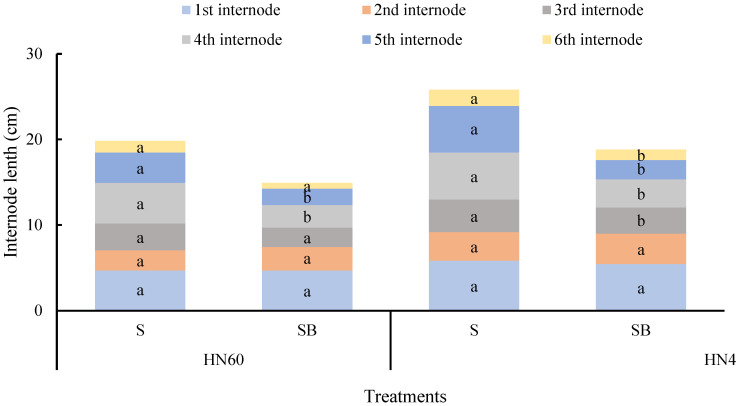
Difference in internode length of soybean stem under blue light treatment and under shading (cm). HN60: Henong 60; HN48: Heinong 48. S: shading treatment; SB: shade + blue light. Means were compared using one-way ANOVA. Different letters in the same internode of each variety in the figure indicate significant differences between treatments (*p* < 0.05, n = 10). The measurements were taken from distinct samples (biological replicates).

**Figure 3 ijms-26-04411-f003:**
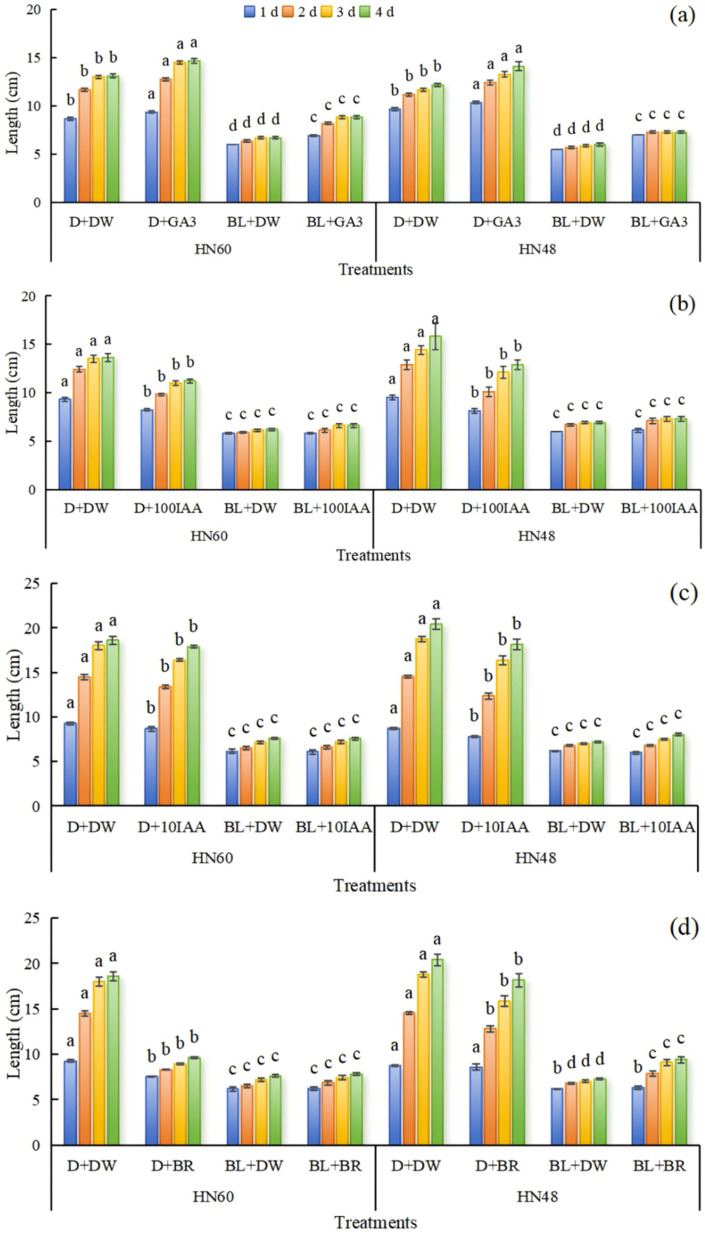
Effects of exogenous GA_3_ (**a**), IAA (**b**,**c**), and BR (**d**) on hypocotyl length of soybean. HN60: Henong 60; HN48: Heinong 48. D + DW: darkness + distilled water; D + GA_3_: darkness + GA_3_; BL + DW: blue light + distilled water; BL + GA_3_: blue light + GA_3_; D + 100IAA: darkness + 100 mg L^−1^ IAA; BL + 100IAA: blue light + 100 mg L^−1^ IAA; D + 10IAA: darkness + 10 mg L^−1^ IAA; BL + 10IAA: blue light + 10 mg L^−1^ IAA; D + BR: darkness + BR; BL + BR: blue light + BR. D + DW: the hypocotyls were smeared with distilled water under dark conditions; D + GA_3_, D + 100IAA, D + 10IAA, and D + BR: the hypocotyls were smeared with 100 mg L^−1^ GA_3_, 100 mg L^−1^ IAA, 10 mg L^−1^ IAA, and 100 mg L^−1^ BR under dark conditions. BL + DW: the hypocotyls were smeared with distilled water under blue light conditions; BL + GA_3_, BL + 100IAA, BL + 10IAA, and BL + BR: the hypocotyls were smeared with 100 mg L^−1^ GA_3_, 100 mg L^−1^ IAA, 10 mg L^−1^ IAA, and 100 mg L^−1^ BR under blue light conditions. Means were compared using one-way ANOVA. Different letters in the same column of each variety on the same day indicated significant differences between treatments (*p* < 0.05, n = 6). The measurements were taken from distinct samples (biological replicates).

**Figure 4 ijms-26-04411-f004:**
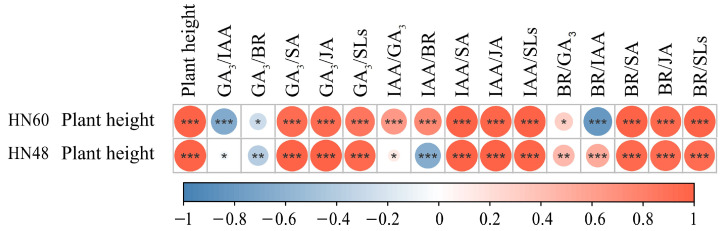
Correlation between plant height and hormone ratio. HN60: Henong 60; HN48: Heinong 48. *: significant difference at the 0.05 level; **: significant difference at the 0.01 level; ***: significant difference at the 0.001 level.

**Figure 5 ijms-26-04411-f005:**
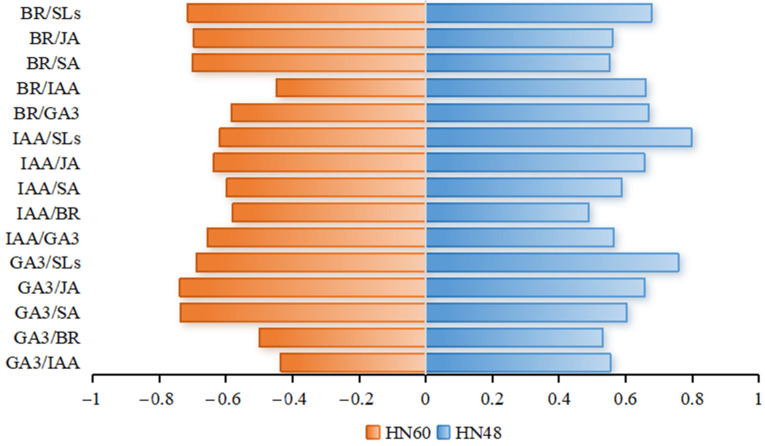
Correlation degree. HN60: Henong 60; HN48: Heinong 48. The number in the figure indicates the correlation degree.

**Table 1 ijms-26-04411-t001:** Differences in hypocotyl length and plant height under blue light treatment in darkness (cm).

Treatments		0 d	1 d	3 d	5 d	7 d	Plant Height
HN60	Darkness	5.13 ± 0.13 a	9.00 ± 0.41 a	14.50 ± 0.65 a	15.25 ± 0.66 a	15.25 ± 0.66 a	36.90 ± 0.29 a
	BL	5.13 ± 0.13 a	5.50 ± 0.00 b	6.00 ± 0.29 b	6.00 ± 0.29 b	6.00 ± 0.29 b	13.28 ± 0.82 b
HN48	Darkness	5.00 ± 0.00 a	9.75 ± 0.32 a	16.25 ± 0.66 a	16.88 ± 0.31 a	16.88 ± 0.31 a	38.18 ± 0.22 a
	BL	5.00 ± 0.00 a	5.25 ± 0.14 b	6.00 ± 0.00 b	6.13 ± 0.13 b	6.13 ± 0.13 b	13.50 ± 0.54 b

HN60: Henong 60; HN48: Heinong 48. BL: blue light treatment. Means were compared using one-way ANOVA. Different letters in the same column of each variety on the same day indicated significant differences between treatments (*p* < 0.05, n = 4). The measurements were taken from distinct samples (biological replicates).

**Table 2 ijms-26-04411-t002:** Difference of stem hormone concentration under blue light treatment and under shading.

Varieties	Treatments		GA_3_ (ng g^−1^)	IAA (ng g^−1^)	BR (ng g^−1^)	JA (ng g^−1^)	SA (μg g^−1^)	SLs (ng g^−1^)
HN60	3rd internode	S	54.77 ± 0.41 a	36.17 ± 0.46 a	7.25 ± 0.09 a	3.81 ± 0.06 b	15.48 ± 0.13 b	107.41 ± 0.46 b
		SB	42.62 ± 0.31 b	28.75 ± 0.65 b	6.08 ± 0.03 b	4.64 ± 0.03 a	17.93 ± 0.20 a	136.13 ± 0.59 a
	4th internode	S	46.28 ± 0.52 a	34.34 ± 0.6 a	6.83 ± 0.06 a	3.62 ± 0.04 b	15.49 ± 0.13 b	109.96 ± 0.16 b
		SB	42.86 ± 0.33 b	28.31 ± 0.08 b	6.02 ± 0.01 b	4.44 ± 0.02 a	17.93 ± 0.20 a	133.47 ± 0.22 a
	5th internode	S	43.78 ± 0.07 a	34.30 ± 0.35 a	7.11 ± 0.04 a	3.58 ± 0.05 b	13.65 ± 0.56 b	110.80 ± 0.28 b
		SB	40.98 ± 0.39 b	26.41 ± 0.29 b	5.81 ± 0.07 b	4.27 ± 0.04 a	17.18 ± 0.03 a	126.12 ± 1.69 a
HN48	3rd internode	S	55.64 ± 0.07 a	46.08 ± 0.58 a	6.32 ± 0.12 a	4.78 ± 0.14 b	10.69 ± 0.24 b	103.39 ± 1.44 b
		SB	46.99 ± 1.27 b	43.30 ± 0.03 b	6.10 ± 0.06 a	7.15 ± 0.07 a	15.03 ± 0.14 a	128.86 ± 0.25 a
	4th internode	S	48.91 ± 0.08 a	45.79 ± 0.48 a	6.98 ± 0.14 a	4.71 ± 0.24 b	9.66 ± 0.09 b	96.49 ± 0.04 b
		SB	44.05 ± 0.08 b	38.52 ± 0.24 b	5.19 ± 0.15 b	7.09 ± 0.03 a	14.10 ± 0.14 a	126.42 ± 0.22 a
	5th internode	S	45.84 ± 0.09 a	44.43 ± 0.33 a	7.08 ± 0.15 a	4.37 ± 0.30 b	9.22 ± 0.46 b	98.93 ± 0.36 b
		SB	41.07 ± 0.48 b	37.26 ± 0.33 b	5.12 ± 0.27 b	6.76 ± 0.11 a	13.10 ± 0.23 a	125.14 ± 2.52 a

HN60: Henong 60; HN48: Heinong 48. S: shading treatment; SB: shade + blue light. GA_3_: gibberellin A3; IAA: indole-3-acetic acid; BR: brassinolide; JA: jasmonic acid; SA: salicylic acid; SLs: strigolactones. Means were compared using one-way ANOVA. The different letters in the same column of the same section of each variety in the table indicate significant differences between treatments (*p* < 0.05, n = 4). The measurements included two biological replicates and two technical replicates.

**Table 3 ijms-26-04411-t003:** Differences in hypocotyl hormone concentrations under blue light treatment in darkness.

Treatments		GA_3_ (ng g^−1^)	IAA (ng g^−1^)	BR (ng g^−1^)	JA (ng g^−1^)	SA (μg g^−1^)	SLs (ng g^−1^)
HN60	Darkness	48.28 ± 0.54 a	55.34 ± 0.86 a	7.50 ± 0.20 a	4.36 ± 0.02 b	9.47 ± 0.04 b	128.11 ± 3.62 b
	BL	41.47 ± 0.69 b	40.71 ± 0.51 b	6.19 ± 0.03 b	5.01 ± 0.09 a	14.80 ± 0.43 a	220.56 ± 0.78 a
HN48	Darkness	54.00 ± 0.84 a	54.87 ± 1.32 a	7.26 ± 0.08 a	4.05 ± 0.02 b	10.82 ± 0.06 b	137.70 ± 1.23 b
	BL	43.71 ± 1.02 b	37.01 ± 0.27 b	5.61 ± 0.23 b	5.58 ± 0.27 a	15.26 ± 0.09 a	215.96 ± 9.48 a

HN60: Henong 60; HN48: Heinong 48. BL: blue light treatment. GA_3_: gibberellin A3; IAA: indole-3-acetic acid; BR: brassinolide; JA: jasmonic acid; SA: salicylic acid; SLs: strigolactones. Means were compared using one-way ANOVA. The different letters in the same column of each variety in the table indicate significant differences between treatments (*p* < 0.05, n = 4). The measurements included two biological replicates and two technical replicates.

**Table 4 ijms-26-04411-t004:** Differential proteins involved in plant hormone metabolism and signal transduction under blue light.

KO(Name)	BL/Dark	*t*-Test *p*-Value	Regulation	Hormones Involved
K14508(NPR1)	1.401	0.001	Up	Salicylic acid
K14493(GID1)	0.816	0.018	Down	Gibberellin
K07513(ACAA1)	1.480	0.001	Up	Jasmonic acid
K15718(LOX1_5)	1.471	0.025	Up	
K15718(LOX1_5)	1.308	0.046	Up	
K00454(LOX2S)	1.254	0.001	Up	
K10528(HPL)	1.371	0.011	Up	
K10525(AOC)	0.740	0.015	Down	
K01723(AOS)	0.788	0.000	Down	
K01723(AOS)	0.772	0.000	Down	
K00128(ALDH)	0.726	0.006	Down	Indole-3-acetic acid
K00128(ALDH)	0.719	0.024	Down	
K14085(ALDH7A1)	0.595	0.000	Down	
K14085(ALDH7A1)	0.550	0.003	Down	

BL: blue light treatment. NPR1: nonexpressor of pathogenesis-related genes 1; GID1: GA-INSENSITIVE DWARF 1; ACAA1: acetyl-CoA acyltransferase 1; LOX1_5: lipoxygenase 1_5; LOX2S: lipoxygenase2S; HPL: hydroperoxide lyase; AOC: allene oxide cyclase; AOS: allene oxide synthase; ALDH: aldehyde dehydrogenase; ALDH7A1: aldehyde dehydrogenase 7A1.

## Data Availability

The data presented in this study are available on request from the corresponding author.

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
