# Peer review of "Hormone Regulation Effect of Blue Light on Soybean Stem Internode Growth Based on the Grey Correlation Analysis Model"

_ijms, 2025, doi:10.3390/ijms26094411_

Round 1
Reviewer 1 Report
Comments and Suggestions for Authors
- The aim of the study is missing in the abstract.
- It would be better when the object of the study is included in the text for the first time, to be included with its full scientific name of the species and after that with its common name: Glycine max (soybean). Once included in the text, you could use the short name of the species written in italics - G. max or its common name, soybean.
- Please, write the scientific names of the species in italics.
- The abbreviations should be defined the first time they appear in each of three sections: the abstract; the main text; the first figure/table. When defined for the first time, the abbreviation should be added in parentheses after the written-out form.
- What are Henong 60 and Heinong 48? The first time they appear in the text, it should be specified that they are varieties. It should be mentioned in the abstract that they are soybean varieties.
- “One-way” ANOVA is with capital letter.
- According to the requirements of the journal, figures/tables should be inserted into the main text close to their first citation. My suggestion is to follow the order: first, mention the figure/table in the text, then the concrete figure/table and then the description of the results presented on this figure/table.
- All figures/tables should have a short explanatory title and caption. What are HN60 and HN48 at figures/tables? Probably are Henong 60 and Heinong 48, but that should be mentioned in the caption of the figures/tables. All the abbreviations on figure 3 should be explained in the caption (D+DW; BL+DW; D+BR, etc.).
- The sentence in line 181 should be corrected with a capital letter.
Author Response
|
Comments 1: The aim of the study is missing in the abstract. |
|
Response 1: Thank you for pointing this out. We agree with this comment. Therefore, we have added the study aim to the abstract( page 1, line11-15). |
|
Comments 2: It would be better when the object of the study is included in the text for the first time, to be included with its full scientific name of the species and after that with its common name: Glycine max (soybean). Once included in the text, you could use the short name of the species written in italics - G. max or its common name, soybean. |
|
Response 2: Thank you for this suggestion. We have included the full scientific name of the species followed by the common name in the abstract and main text where the species is first mentioned (page 1, line 12; page 1, line 36; page 2, line 46). |
|
Comments 3: Please, write the scientific names of the species in italics. |
|
Response 3: We acknowledge this requirement. All scientific names of species (e.g., Glycine max) have been italicized throughout the manuscript (page 1, line 12, line36; page 2, line 46). |
|
Comments 4: The abbreviations should be defined the first time they appear in each of three sections: the abstract; the main text; the first figure/table. When defined for the first time, the abbreviation should be added in parentheses after the written-out form. |
|
Response 4: Thank you for highlighting this. Abbreviations have been defined in parentheses at their first occurrence in the abstract, main text, and figures/tables.(page 1, line 23-25; page 2, line 20、55、56、62、66、84; page 3, line97-98,104-105; page 4, line113、126; page 5, line140-142、153-154; page 7, line172-176; page 8, line190-195、199-202; page 9, line212、223; page 10, line238-239、250、253-254、261、274; page 11, line290; page 13, line397) |
|
Comments 5: What are Henong 60 and Heinong 48? The first time they appear in the text, it should be specified that they are varieties. It should be mentioned in the abstract that they are soybean varieties. |
|
Response 5: Henong 60 and Heinong 48 are soybean varieties. We have added this clarification in the abstract and main text.(page 1, line 16-17; page 2, line 83-84) |
|
Comments 6: “One-way” ANOVA is with capital letter. |
|
Response 6: We apologize for the oversight. "One-way" ANOVA has been corrected with capital letters (page 4, line 114、127; page 5, line 142、155; page 8, line 182) |
|
Comments 7: According to the requirements of the journal, figures/tables should be inserted into the main text close to their first citation. My suggestion is to follow the order: first, mention the figure/table in the text, then the concrete figure/table and then the description of the results presented on this figure/table. |
|
Response 7: We have revised the manuscript to insert figures/tables close to their first citation. All other figures/tables have been positioned close to their first citation as required.(page 3, line106-110) |
|
Comments 8: 8.All figures/tables should have a short explanatory title and caption. What are HN60 and HN48 at figures/tables? Probably are Henong 60 and Heinong 48, but that should be mentioned in the caption of the figures/tables. All the abbreviations on figure 3 should be explained in the caption (D+DW; BL+DW; D+BR, etc.). |
|
Response 8: All figures/tables now have short explanatory titles and captions(Figure3: page 7, line173-176). HN60 and HN48 are defined as Henong 60 and Heinong 48 in all figure/table captions(page 1, line 16-17; page 2, line 83-84; page 3, line104-105; page 4, line113、126; page 5, line140、153; page 7, line173; page 9, line212、223). |
|
Comments 9: The sentence in line 181 should be corrected with a capital letter. |
|
Response 9: Thank you for pointing this out. During the manuscript revision, we rewrote the sentence in line 181 for clarity and ensured all sentences start with a capital letter. Additionally, we conducted a full proofread to confirm all sentences in the manuscript adhere to capitalization rules. |
Reviewer 2 Report
Comments and Suggestions for Authors
Re: ijms-3509908
Wang et al. presented a research article entitled “Hormone regulation effect of blue light on soybean stem internode growth based on the grey correlation analysis model” for publication consideration in IJMS.
The authors conducted a study on how blue light serves as a crucial environmental signal influencing the morphological regulation of soybean stem internode growth, employing a grey correlation analysis model. The article is scientifically sound and well presented. Three sets of experiments were designed, and the authors suggest that the ratio of growth-promoting to growth-inhibiting hormones may be a key factor contributing to the morphological differences observed between the two varieties.
It would be appreciated if the authors could refine the abstract to enhance its accuracy and formatting, as the current version reads more like a proposal report. Additionally, please ensure the manuscript’s overall readability and logical flow before publication.
Comments on the Quality of English LanguagePlease make sure the readiness and flow of the manuscript before publication.
Author Response
|
Comments 1: It would be appreciated if the authors could refine the abstract to enhance its accuracy and formatting, as the current version reads more like a proposal report. |
|
Response 1: Thank you for this suggestion. We agree that the abstract should be refined to better reflect a research article. Therefore, we have restructured the abstract to include background, methods, results, and conclusions (page 1, lines 11-37). |
|
Comments 2:please ensure the manuscript’s overall readability and logical flow before publication. |
|
Response 2: Thank you for emphasizing the importance of readability and logical flow. We have conducted a comprehensive revision and polishing of the entire manuscript to enhance clarity and coherence. |
|
2. Response to Comments on the Quality of English Language |
|
Response 1:We have conducted a comprehensive revision and polishing of the entire manuscript. |